# Pseudointelligence:
# A Unifying Framework for Language Model Evaluation

**Shikhar Murty**[*]
Stanford University
smurty@cs.stanford.edu

**Orr Paradise**[*]
UC Berkeley
orrp@eecs.berkeley.edu

**Pratyusha Sharma**[*]
MIT
pratyusha@mit.edu

## Abstract

With large language models surpassing human performance on an increasing number of benchmarks, we must take a principled approach for targeted evaluation of model capabilities. Inspired by pseudorandomness, we propose *pseudointelligence*, which captures the maxim that "(perceived) intelligence lies in the eye of the beholder." That is, that claims of intelligence are meaningful only when their evaluator is taken into account. Concretely, we propose a complexity-theoretic framework of model evaluation cast as a dynamic interaction between a model and a learned evaluator. We demonstrate that this framework can be used to reason about two case studies in language model evaluation, as well as analyze existing evaluation methods.

## 1 Introduction

Recent works claim that GPT-4 achieves expert-level performance on complex reasoning tasks (Katz et al., 2023; Lin et al., 2023), with some researchers concluding that it exhibits sparks of intelligence (Bubeck et al., 2023).

But how should intelligence be evaluated? This question dates back to Descartes (1637), formalized by Turing (1950), and continues to be the subject of recent discussion (Chollet 2019; Mitchell and Krakauer 2023; Burnell et al. 2023 *inter alia*). However, none of these attempts prescribe a particular evaluator (e.g., sequence of questions) that guarantees the intelligence of the evaluated model.

This is not a coincidence. We argue that intelligence is in the eye of the evaluator. This maxim is particularly important for the future of natural language processing (NLP): progress cannot be measured by static benchmarks (Raji et al., 2021; Hutchinson et al., 2022; Shirali et al., 2023), with contemporary models surpassing human performance on new evaluations within a few years (Kiela

---

[*]Equal contribution. Authors listed alphabetically.

et al., 2021), and benchmarks leaking into training data (Elangovan et al., 2021).

Instead, we define the notion of *pseudointelligence*. Analogous to pseudorandomness (Blum and Micali, 1984; Yao, 1982), which measures a distribution by its distinguishability from true randomness, pseudointelligence applies to the evaluation of the capabilities of learned models. Importantly, a claim that a model has learned a certain capability is innately entangled with the distinguishing ability of an evaluator.

With the future of NLP in mind, we focus on *learned evaluators*. These evaluators are trained on samples specific to a given capability, much like the models they assess. Notably, emerging evaluation methods, such as model-based evaluation (Perez et al., 2023; Ribeiro et al., 2021) and adversarial evaluation (Jia and Liang, 2017; Nie et al., 2020; Bartolo et al., 2020), can be viewed as specific instances of the framework we propose. Our main takeaways are:

**P1:** A claim of intelligence must be supplemented by an explicitly-defined *evaluator* and (intelligent) *capabilities* (Section 3.1).

**P2:** Increased resources dedicated to model development should be accompanied by *increased resources dedicated to evaluation*. These include the number of examples of the capability, and the complexity of the space of possible models and evaluators (Section 3.2).

**P3:** *Self-evaluation* cannot support a claim of intelligence if the evaluator is directly derived from the model. It might, however, be useful as means towards a different end (Section 3.3).

Besides laying the foundation for theoretical analysis, our framework also provides a *unifying lens* on existing evaluation methods (Section 4).

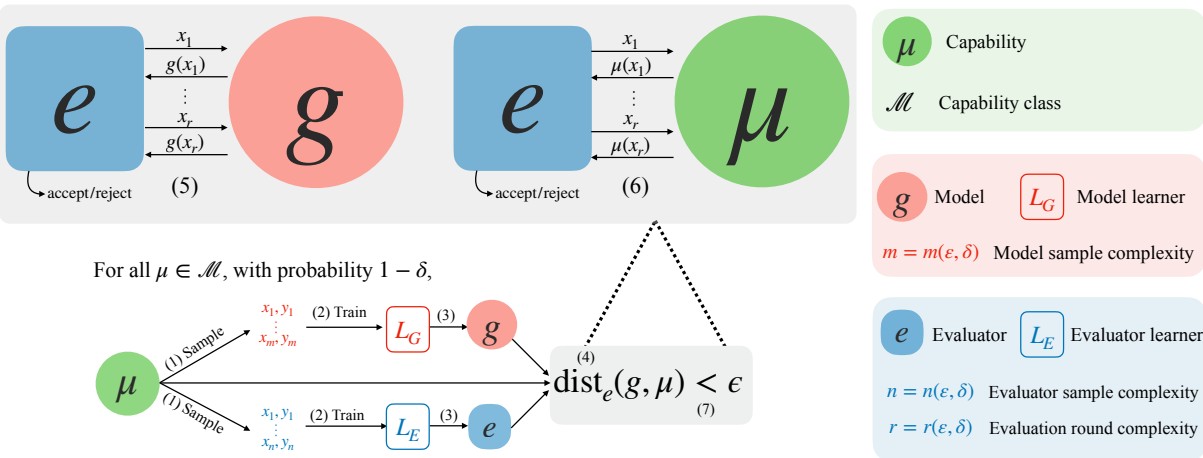

Figure 1: *Targeted evaluation of a pseudointelligent model.* For each capability $\mu$, (1) iid samples are drawn and (2) fed to the learners, which (3) output a model and an evaluator. (4) The distinction $\text{dist}_e(g, \mu)$ is computed as the expected difference in evaluator output during a multi-round interaction with (5) the model $g$ versus (6) the ground-truth capability $\mu$. (7) If $\text{dist}_e(g, \mu) < \varepsilon$ with probability[1] greater than $(1 - \delta)$, we say that $L_G$ is pseudointelligent against $L_E$ w.r.t capabilities $\mathcal{M}$. See Definition 3.2 for a formal definition. Note that the targeted evaluator is trained on samples from the capability $\mu$, and adaptively interacts with the model $g$.

## 2 Background

**Pseudorandomness.** First, a brief introduction of pseudorandomness, which forms the conceptual backbone of our framework. For an extended introduction, see Goldreich (2008).

Tessa and Obi are playing a game, and would like to decide who gets to go first. They agree to make the decision based on a coin toss: Tessa tosses a coin, and Obi calls *Heads* or *Tails*. If Obi calls the outcome correctly he gets to go first, and otherwise Tessa does. Now consider two cases:

1. Obi is calling the coin based only on the information available to him from eyesight.

2. Obi has access to an array of sensors that measure the initial conditions of Tessa's coin toss, and a powerful computer that can perform complicated calculations in a millisecond.

Tessa would not be happy with a coin toss in the second case, because Obi could call the coin correctly with ease. In other words, the coin toss is no longer "random-enough" due to Obi's increased computational power. More generally, a distribution is *pseudorandom against a particular observer* if she cannot distinguish it from a truly random. Formally,

**Definition 2.1.** Fix $\varepsilon \in (0, 1)$ and a finite set $\mathcal{X}$. Let $\mathcal{U}_\mathcal{X}$ denote the uniform distribution over $\mathcal{X}$. A

distribution $\mathcal{P}$ over $\mathcal{X}$ is $\varepsilon$-pseudorandom against a class of distinguishers $D$ if for every $d \in D$,

$$\left| \Pr_{x \leftarrow \mathcal{P}} [d(x) \text{ accepts}] - \Pr_{x \leftarrow \mathcal{U}_\mathcal{X}} [d(x) \text{ accepts}] \right|$$
$$< \varepsilon.$$

One can view Definition 2.1 as consisting of an *ideal* source (uniformly random elements), and a *pseudoideal* approximation to this source (pseudorandom elements). Unlike randomness, intelligence does not have a canonical mathematical operationalization.

**The Turing Test.** In the Turing Test (Turing, 1950), an evaluator converses with either a machine or a human; the machine attempts to convince the evaluator that it is human, while the evaluator aims to distinguish machine from human. If the machine successfully fools the evaluator, Turing argued that it should be considered as exhibiting intelligent behavior. However, while passing the Turing test signifies that the machine is indistinguishable from human by a particular evaluator, it alone does not imply human-level learning or comprehension (independent of an evaluator). Pseudointelligence is defined with this intuition in mind; however, it *explicitly* requires specifying the particular evaluator and (intelligent) capabilities against which the machine is measured.

---

[1] Over the samples from $\mu$, and any randomness used by the learners $L_G$, $L_E$, model $g$ and evaluator $e$.

## 3 Pseudointelligence

Our main message (**P1**) is that claims of intelligence should center the evaluator, and not just the (allegedly) intelligent model. Put differently, a claim that a model is intelligent is actually a claim that it is "intelligent-enough," therefore it is meaningful only with respect to a specific class of evaluators. We provide a complexity-theoretic framework in which evaluators are placed front and center, formalizing Figure 1.

### 3.1 Setup

A *model* is a (possibly randomized) mapping $g\colon \mathcal{X} \to \mathcal{Y}$, where $\mathcal{X}$ is a set of *queries* and $\mathcal{Y}$ is a set of responses.

**Capabilities.** A *capability* is a distribution $\mu$ over $\mathcal{X} \times \mathcal{Y}$. For a given query $x \in \mathcal{X}$, we let $\mu(x)$ denote a sample from the conditional distribution on acceptable responses $\mu(\cdot \mid x)$; thus, $\mu$ can be thought of as the ground-truth randomized mapping $\mu\colon \mathcal{X} \to \mathcal{Y}$ against which models are evaluated.

**Evaluators.** In this work, we study the *perceived intelligence* of a model. That is, how well a model appears to posses certain capabilities *as perceived by an evaluator.*[2] We formalize this by considering an evaluator $e$ which is an algorithm that is given black-box access to the model $g$; for each of $i \in [r]$ rounds, the evaluator queries $g$ on $x_i$ to receive response $y_i$; finally, the evaluator "accepts" $g$ if it thinks it is the ground-truth capability, and rejects it otherwise. Note that the query $x_i$ may depend on previous responses $y_1, \ldots, y_{i-1}$.

The degree to which an evaluator $e$ is able to distinguish between the model $g$ and a (ground-truth) capability $\mu$ is defined next.

**Definition 3.1** (Distinction)**.** Let $e$ be an evaluator, $g\colon \mathcal{X} \to \mathcal{Y}$ be a model and $\mu$ be a capability over $\mathcal{X} \times \mathcal{Y}$. For any $\varepsilon \in (0, 1)$, we say that $e$ can $\varepsilon$-*distinguish* between $g$ and $\mu$ if

$$\underbrace{|\Pr\left[e \text{ accepts } g\right] - \Pr\left[e \text{ accepts } \mu\right]|}_{\text{dist}_e\,(g,\mu)} > \varepsilon.$$

If $\text{dist}_e\,(g,\mu) \leq \varepsilon$ then we say that $e$ *cannot* $\varepsilon$-*distinguish* between $g$ and $\mu$.

The distinction $\text{dist}_e\,(g,\mu)$ captures the likelihood that an evaluator distinguishes a *given* model

---

$g$ from the (ground-truth) capability $\mu$. However, intelligence is not the same as possessing a particular capability (Gunderson and Gunderson, 2008). Rather, we view it as an ability to *learn* various capabilities. Thus, we consider a learner $L_G$ that learns a model $g \in G$ from finite samples of $\mu$.

We will say that the learner is pseudointelligent if, with high probability, the evaluator cannot distinguish between the learned model and the capability. Lastly, to allow for *targeted evaluation* of the capability, we consider an evaluator learner $L_E$ that is also given (different) samples from the capability, and outputs an evaluator $e \in E$ targeted at it.

**Definition 3.2** (Pseudointelligence)**.** Fix a query set $\mathcal{X}$, response set $\mathcal{Y}$, and a class of capabilities $\mathcal{M}$. Fix sample complexity functions $m, n\colon (0, 1)^2 \to \mathbb{N}$. Given a model class $\mathcal{G} = (G, L_G, m)$ and an evaluator class $\mathcal{E} = (E, L_E, n)$, we say that $\mathcal{G}$ is *pseudointelligent* with respect to $\mathcal{E}$ and capabilities $\mathcal{M}$ if, for any $\varepsilon, \delta \in (0, 1)$, whenever $L_G$ (resp. $L_E$) is given $m := m(\varepsilon, \delta)$ (resp. $n := n(\varepsilon, \delta)$) iid samples from $\mu$, with probability at least $1 - \delta$,[3] $L_G$ and $L_E$ output model $g$ and evaluator $e$ such that $e$ cannot $\varepsilon$-distinguish between $g$ and $\mu$:

$$\forall \mu \in \mathcal{M} \Pr_{\substack{g \leftarrow L_G \circ \mu^m \\ e \leftarrow L_E \circ \mu^n}} [\text{dist}_e(g, \mu) \leq \varepsilon] \geq 1 - \delta.$$

Note that the number of rounds of interaction between the evaluator $e$ and the model $g$ (denoted $r := r(\varepsilon, \delta)$ in Figure 1), also scales with $\varepsilon$ and $\delta$. Next, we examine two case-studies to understand the effect of the implicit parameters in Definition 3.2 on the validity of claims of intelligence.

### 3.2 Model resources vs. evaluator resources

Our main message (**P2**) underscores the importance of resources allocated to the evaluator relative to those allocated to the model. There are several axes on which this comparison can be made:

**Samples.** To evaluate capabilities $\mathcal{M}$ within error $\delta$ and distinction $\varepsilon$, the model learner is given $m(\varepsilon, \delta)$ samples and the evaluator learner is given $n(\varepsilon, \delta)$ samples of each capability $\mu \in \mathcal{M}$. How do each of these grow as a function of $\delta$ and $\epsilon$?

**Learner expressivity.** The model learner $L_G$ outputs a model $g \in G$, and the evaluator learner $L_E$ outputs an evaluator $e \in E$. How expressive is the class of possible models $G$ as compared to the

---

[2]We prefer *evaluator* over *benchmark* as it emphasizes its role as an active participant in an interaction, rather than a passive dataset.

[3]*Ibid.*, 1.

class of possible evaluators $E$? A naive measure of expressivity compares the number of parameters needed to encode each: $\log |G|$ vs. $\log |E|$. Supervised learning theory has more refined measures that can be applied to infinite spaces and provide tighter bounds (Natarajan, 1989; Daniely and Shalev-Shwartz, 2014). While these measures can be applied to the model class, new measures must be developed to capture evaluator classes.

**Learner compute resources.** How much computational power is used to train $L_G$ and $L_E$? Note that learner expressivity is concerned only with the existence of a model $g \in G$ that is indistinguishable by the evaluator, but not with how to find it. This search takes compute resources; the amount of resources available to $L_G$ vs. $L_E$ affects the outcome of the evaluation.

**Model and evaluator computational power.** Given a query $x \in \mathcal{X}$, how much computational power is needed to compute a response $g(x)$? On the evaluator side, how much power is needed to compute the $i$th query issued by the evaluator, given the preceding $(i - 1)$ queries and responses? Additionally, given a full evaluation $(x_i, y_i)_{i=1}^r$, how much power is needed by the evaluator to decide whether it accepts?

### 3.3 Should a model evaluate itself?

One particularly interesting case is when the model is pitted against itself by playing a dual rule: both model *and* evaluator. Self-evaluation can be used to assist human evaluators (Saunders et al., 2022) or to increase model "honesty" (Kadavath et al., 2022). The validity of self-evaluation for claims of intelligence remains contested (cf. Zhang et al., 2023 and the discussion around it), and is the focus of this case study.

To consider self-evaluation in our framework, we first map models onto evaluators $g \mapsto e_g$.[4] Once such a mapping is fixed, we map a model learner $L_G$ to an evaluator learner $L_{E_G}$ that, given samples $S \leftarrow \mu^n$, computes $g \leftarrow L_G(S)$ and outputs $e_g$.

Can $L_G$ be pseudointelligent with respect to $L_{E_G}$? This is akin to asking whether $L_G$ is pseudointelligent *with respect to itself*. This brings us to a crucial detail of our framework: For self-evaluation to fit in our framework, $L_{E_G}$ and $L_G$

---

[4] For example, consider the case that $g$ models yes-no questions ($\mathcal{Y} = \{0, 1\}$). Then one can obtain an evaluator $e_g$ from a model $g$ by sampling a query $x$, querying the black box to receive a response $y$, and accepting if and only if $g(x) = y$.

should receive *independent* samples from $\mu$. This is in stark contrast to the existing practice of deriving the evaluator directly from the trained model $\hat{g} \mapsto e_{\hat{g}}$ (Kadavath et al., 2022; Saunders et al., 2022; Zhang et al., 2023). Our main message (**P3**) is that this does *not* show that $L_G$ is pseudointelligent—although it may be useful as means towards a different end, as in Kadavath et al. (2022); Saunders et al. (2022).

## 4 Existing evaluations through the lens of pseudointelligence

Pseudointelligence can serve a *unifying* role by allowing a direct comparison between different evaluation methods. We cast several existing evaluation paradigms into our framework.

**Static Datasets.** The evaluator memorizes samples drawn from the capability, and queries its black box on a random sample: Given samples $S \leftarrow \mu^n$, $L_E$ outputs an evaluator $e_S$ that draws a sample $(x, y) \leftarrow S$ at random, queries the black box on $x$, and accepts if and only if the response was $y$. Clearly, like all inductive inference settings, an evaluator can be fooled by any pseudo-intelligent model that just happens to get the correct labels by learning simple shortcuts.

**Adversarial Evaluation (AE).** AE requires access to some *auxiliary model* $\hat{g}$ that $L_E$ can use to search for a challenge test set, which can then be used by an evaluator. Concretely, given seed samples $S$ and an auxiliary model $\hat{g}$, $L_E$ filters out all examples where $\hat{g}$ outputs the correct response, thereby creating a challenge test set $\hat{S}$. Such a filtering process can be done in several rounds, where human annotators modify an initial query until $\hat{g}$ makes an error (Bartolo et al., 2022). Intuitively, based on the quality of $\hat{g}$, such filtering can create increasingly hard datasets. Thus, the central *resources* here are the amount of seed samples $S$ and the complexity of the auxiliary model $\hat{g}$.

**Model-based Evaluation.** These evaluators also use an auxiliary $\hat{g}$, albeit in a non-adversarial way. For instance, Ribeiro et al. (2021) use human-generated templates, filled in by a language model, as queries. Perez et al. (2023) use two auxiliary models: one to generate queries, and the other to find those targeted at a particular capability.

# 5 Conclusion

This paper introduces a principled framework for model evaluation, inspired by the theory of pseudorandomness. Our main message is that claims about model capability must be supplemented with a thorough discussion of the *resources* given to the evaluator, especially in settings where model resources are largely unknown (e.g. OpenAI, 2023). Central to our framework is a model-based evaluator that is targeted at specific capabilities as well as specific models (via multi-round interactions). We hope our framework encourages rigorous analysis of LLM evaluation, and helps unify the study of this increasingly-important topic.

# 6 Limitations

This paper is focused on motivating and defining pseudointelligence, as well as demonstrating its potential use for unifying and analysing LLM evaluation. Deeper analyses, such as provable bounds comparing model and evaluator sample complexities ($m$ vs. $n$), are left for future work.

The impact of large language models extends far beyond their alleged (pseudo-)intelligence (Bommasani et al., 2021). Pseudointelligence does not, for example, correspond to an ability to respond to queries in an ethical or responsible manner. In general, psueodintelligence is concerned with the distinguishing ability of a class of evaluators, but does not consider the usage of a model in a real-world context which may not conform to this class (cf. Mitchell et al., 2019; Suresh et al., 2023). Finally, like all abstract definitions, it must not be used as a *rubber stamp*; that is, it cannot replace a case-by-case assessment of potential impacts of models prior to their deployment.

## Acknowledgements

SM is funded by a gift from Apple Inc. OP is funded by the Simons Collaboration on the Theory of Algorithmic Fairness. PS and OP are funded by Project CETI via grants from Dalio Philanthropies and Ocean X; Sea Grape Foundation; Rosamund Zander/Hansjorg Wyss, Chris Anderson/Jacqueline Novogratz through The Audacious Project: a collaborative funding initiative housed at TED.

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
