# OpenReview forum: "Pseudointelligence: A Unifying Lens on Language Model Evaluation"
_EMNLP/2023/Conference — EMNLP 2023 Findings_

### Official Review · Reviewer_WYM6 · 2023-07-23

**Soundness:** 2

**Excitement:**

2: Mediocre: This paper makes marginal contributions (vs non-contemporaneous work), so I would rather not see it in the conference.

**Paper Topic And Main Contributions:**

The authors introduce the concept of pseudo-intelligence, which resembles the definition of pseudo-randomness. Roughly speaking, a learning algorithm is pseudo-intelligent if it produces a model that cannot be distinguished from a real expert under the limited resources of the distinguisher.

**Reasons To Accept:**

An attempt was made to formalize the evaluation of modern AI models, such as LLMs.

**Reasons To Reject:**

- The work is too preliminary. Algorithms for learning a class of models and a class of evaluators have not been proposed. The conditions under which learning in the proposed framework is successful have not been studied.
- Without instantiating a pseudo-intelligence with a non-trivial evaluator class, the proposed framework remains too general to be useful for theoretical analysis. This is confirmed by the example of a trivial instance ($L_E\equiv e_0$, $e_0\in E$), which is considered in the discussion below.
- Under additional assumptions, the definition of pseudo-intelligence can be reduced to the definition of PAC learnability of the class $G\times E$ (see below).

**Reproducibility:**

N/A: Doesn't apply, since the paper does not include empirical results.

**Reviewer Confidence:**

5: Positive that my evaluation is correct. I read the paper very carefully and I am very familiar with related work.

---

> ### Author Rebuttal · Authors · 2023-08-29
>
> > “Roughly speaking, pseudo-intelligence is a learning algorithm that produces a model that cannot be distinguished from a real expert under the limited computing power of the distinguisher.”
>
> Answer: Pseudo-intelligence is not a learning algorithm. Rather, as defined in the paper, a learner is said to be pseudo-intelligent with respect to an evaluator class if it satisfies certain conditions. We refer to lines 182-193 in the paper for the full definition.
>
> > The work is too preliminary. Algorithms for learning a class of models and a class of evaluators have not been proposed. The conditions under which learning in the proposed framework is successful have not been studied.
>
> Answer: As a theme track paper, we introduce a framework that highlights the importance of providing sufficient _resources_ to the evaluator, and a sketch for how these evaluators may be constructed. The framework is not concerned with the learning of models, and we leave practical implementations of such evaluators to future non-theme track papers.
>
> > The advantage of the proposed framework compared to Valiant's PAC learning framework has not been studied.
>
> Answer: PAC Learning is not comparable to our framework. For instance, PAC learning is concerned with the ability of a learner to output a hypothesis that generalizes to any distribution $\mu$; Futhermore, the error is measured with respect to this distribution $\mu$. However, the error of a pseudointelligent learner is measured by a learned evaluator and is defined with respect to a particular capability (distribution). In other words, PAC learning only has a learner, whereas pseudointelligence is defined with respect to a learner, a (learned) evaluator, and a capability.

---

### Official Review · Reviewer_Mo8o · 2023-08-05

**Soundness:** 3

**Excitement:**

4: Strong: This paper deepens the understanding of some phenomenon or lowers the barriers to an existing research direction.

**Paper Topic And Main Contributions:**

This paper is about how should we evaluate the intelligence of large language models, especially when they perform at or above human levels on certain tasks?
The paper introduces the concept of "pseudointelligence",  suggesting that perceived intelligence is subjective and depends on the evaluator's perspective.
Inspired by pseudorandomness which measures a distribution by its distinguishability from true randomness, pseudointelligence evaluates the capabilities of learned models based on their distinguishability from certain capabilities.
The authors propose a framework where model evaluation is seen as a dynamic interaction between the model and a learned evaluator, emphasizing that claims of a model's intelligence are meaningful only when considering the evaluator.
The paper uses this framework to reason about two case studies in language model evaluation. It also analyzes existing evaluation methods under this new lens.
This work conveys three main messasges: (1) Claims of intelligence must be supplemented by a clearly defined evaluator and its capabilities. (2) As more resources are dedicated to model development, there should also be an increase in resources dedicated to evaluation. (3) Self-evaluation by a model cannot support a claim of intelligence if the evaluator is derived directly from the model itself.

**Questions For The Authors:**

1. Could you provide more examples or case studies where this concept can be applied to evaluate the capabilities of large language models?

2. Are there plans to conduct more empirical studies to validate the proposed framework?

3. The paper introduces complex concepts like pseudointelligence and pseudorandomness. Could you provide more explanation or examples to make these concepts accessible to a broader audience?

4. While the paper presents a meticulous theoretical framework, it seems to lack empirical validation. Are there plans to conduct empirical studies to validate the proposed framework?

**Reasons To Accept:**

1. This paper provides a fresh perspective on evaluating the capabilities of large language models.

2. This paper present a meticulous and detailed theoretical framework for understanding and evaluating pseudointelligence, providing a solid foundation for future research and discussions in this area.

3. The 3 messages are reasonable, and they are also inspiring to related topics.

**Reasons To Reject:**

1. The paper introduces a novel concept of pseudointelligence and a framework for evaluating language models. However, it might lack empirical validation or case studies that demonstrate the practical utility of this framework.

2. The paper primarily focuses on the theoretical aspect of pseudointelligence. It might not provide a comprehensive discussion on how this concept can be applied to various NLP tasks or different types of language models.

3. The paper introduces complex concepts like pseudointelligence and pseudorandomness. It might not provide enough explanation or examples to make these concepts accessible to a broader audience.

4. The paper might not provide a thorough comparison of the proposed evaluation framework with existing methods. Such a comparison would help readers understand the advantages and potential limitations of the proposed approach.

**Reproducibility:**

N/A: Doesn't apply, since the paper does not include empirical results.

**Reviewer Confidence:**

4: Quite sure. I tried to check the important points carefully. It's unlikely, though conceivable, that I missed something that should affect my ratings.

---

> ### Author Rebuttal · Authors · 2023-08-29
>
> Thank you for your thoughtful and helpful review. We are glad you found the paper meticulous and inspiring.
>
> > The paper introduces a novel concept of pseudointelligence and a framework for evaluating language models. However, it might lack empirical validation or case studies that demonstrate the practical utility of this framework.
>
> Answer: The main utility of this framework is a unifying, theoretical lens on evaluation. In Section 4, we present case studies to cast existing evaluations under the framework of pseudo-intelligence. Since this is a theme track paper at 4 pages, the space was used to convey the main idea. We leave empirical validation/implementation for a future non-theme track paper.
>
> > The paper primarily focuses on the theoretical aspect of pseudointelligence. It might not provide a comprehensive discussion on how this concept can be applied to various NLP tasks or different types of language models.
>
> Answer: We provide a sketch of how this framework can be applied to current evaluation methods in Section 4.  Due to space constraints, we avoided directly quoting from each of our cited works with details of how they can be instantiated by our framework.
>
> More generally, we do not prescribe a fixed measure of the complexity of the model class which, depending on the model/method/optimizer, could be highly variable. We envision studying attributes such as the computational complexity of the learner/evaluator: run time, size of the network, and number of training samples. Similarly, practitioners might analyze measures such as the number of flops required to train the learner in a particular implementation, monetary cost or run time at inference.
>
> > The paper introduces complex concepts like pseudointelligence and pseudorandomness. It might not provide enough explanation or examples to make these concepts accessible to a broader audience.
>
> Answer: We are happy to provide more background in a future version with the extra page, and add an extended exposition of technical background in the appendix.
>
> > The paper might not provide a thorough comparison of the proposed evaluation framework with existing methods. Such a comparison would help readers understand the advantages and potential limitations of the proposed approach.
>
> Answer: Pseudo-intelligence is a meta-evaluation framework, that is, it can be used to understand the validity of any evaluator. It is, therefore, not a fixed “approach” that requires comparison but a “framework”. Another such framework is the Turing test, which we do compare to in Section 2. We are happy to discuss other relevant frameworks during the discussion period.
>
> > Could you provide more examples or case studies where this concept can be applied to evaluate the capabilities of large language models? Are there plans to conduct more empirical studies to validate the proposed framework?
>
> Answer: This initial paper introduces the framework and is focused on the conceptual and definitional side. Our next step is an empirical paper that is guided by this framework. We will conduct a meta-evaluation of LLaMa similar to the work of Zheng et al. (2023) [1], which uses language models as evaluators on different Machine Translation tasks, but increasing the power of the evaluator; we hypothesize that a learned evaluator increasingly distinguishes between LLaMa and samples from MT Bench as the complexity of the evaluator increases (measured by number of samples, and number of parameters).
> A main benefit of our framework is its ability to augment empirical understandings with theoretical analysis. Since the submission of this paper, we have already proven preliminary scaling theorems that demonstrate how, in theory, the sample complexity of the evaluator-learner $L_E$ must grow with the sample complexity of the model-learner $L_G$. In other words, this theorem shows how many samples are needed for the model to be distinguished from the ground-truth capability. This theorem can indeed be verified experimentally in a setup like the one described in the previous paragraph.
>
> [1] Judging LLM-as-a-judge with MT-Bench and Chatbot Arena. Lianmin Zheng, Wei-Lin Chiang, Ying Sheng, Siyuan Zhuang, Zhanghao Wu, Yonghao Zhuang, Zi Lin, Zhuohan Li, Dacheng Li, Eric. P Xing, Hao Zhang, Joseph E. Gonzalez, Ion Stoica. arXiv, 2023.

---

### Official Review · Reviewer_6eqq · 2023-08-05

**Typos Grammar Style And Presentation Improvements:** 191
**Soundness:** 4

**Excitement:**

4: Strong: This paper deepens the understanding of some phenomenon or lowers the barriers to an existing research direction.

**Missing References:**

This one may be relevant:
Hutchinson et al. Evaluation Gaps in Machine Learning Practice. FAccT '22. https://dl.acm.org/doi/fullHtml/10.1145/3531146.3533233

**Paper Topic And Main Contributions:**

- This thought-provoking position paper proposes a conceptual framework for evaluating LLMs as they surpass human performance in many tasks.
- The central concept is named "pseudo-intelligence", which posits that the model should produce outputs that are $(\epsilon, \delta)$-indistinguishable from those produced by a specific *capability* according to a specific *evaluator*
- The paper casts model evaluation as dynamic interaction between the model and a separately learned evaluator, and introduces a mathematical language based on learning theory which provides the formal definitions for "pseudo-intelligence"
- The authors further advocate for considering *budget of evaluator* in addition to *budget of the model* during evaluation, and also argue that self-evaluation cannot prove intelligence
- Using this framework, the authors describe various existing approaches such as datasets, adversarial evaluation, and model-based evaluation.

**Questions For The Authors:**

A: How would you apply this pseudointelligence framework to existing evaluation practices? Are there any specific recommendations or tools for implementing this approach?

B: What criteria would your propose for designing effective evaluators based on this framework? How would you prevent artifacts like evaluators favoring outputs closer to *its* training distribution?


**Reasons To Accept:**

- A well-motivated conceptual contribution to LLM evaluation, especially as the current landscape gets really messy with models evaluating itself, data leak, etc.
- The learning-theory-motivated mathematical language is nice, and clearly defines many dimensions to consider during evlauation.
- Release of this position paper may encourage discussion around more rigorous evaluation practices.
- The paper is clearly written and presented.

**Reasons To Reject:**

- Not a lot of specifics for how the proposed framework can be implemented to produce a superior evaluation approach
- Discussion of existing evaluation approaches merely describes them in terms of the framework, but does not derive much insights into how optimal each approach is

**Reproducibility:**

N/A: Doesn't apply, since the paper does not include empirical results.

**Reviewer Confidence:**

4: Quite sure. I tried to check the important points carefully. It's unlikely, though conceivable, that I missed something that should affect my ratings.

---

> ### Author Rebuttal · Authors · 2023-08-29
>
> Thank you for your very thoughtful and comprehensive review. We are glad you found this work thought-provoking and a well-motivated conceptual contribution.
>
> > Not a lot of specifics for how the proposed framework can be implemented to produce a superior evaluation approach
>
> Answer: Thank you for the question. Since this is a theme track paper with 4 pages, the space was used to convey the main idea and connect it to existing evaluation methods. Developing a superior evaluation approach based on the proposed framework is left for future work. However, our paper provides guidelines for building such an approach, as conveyed by our main messages P1-P3 (lines 58-70).
>
> > Discussion of existing evaluation approaches merely describes them in terms of the framework, but does not derive much insights into how optimal each approach is
>
> Answer: The optimality of the evaluator is a function of its resources: compute / data annotation budget, time constraints, and model size (if the evaluator is model-based). We make the following observations on the power of different evaluation approaches based on our definition alone: First, it is clear that the distinction of the evaluator can only grow with its round complexity $r$ (because subsequent queries can always be ignored). This verifies the intuition that adaptive evaluation is more powerful than static evaluation. Out of the existing types of evaluation, IID static datasets are less powerful as compared to adversarial evaluation, which is less powerful than dynamic model-based evaluation/or model-based evaluators, using simpler models is less optimal than using expressive models.
>
> ### Regarding the Questions For The Authors
>
> > How would you apply this pseudointelligence framework to existing evaluation practices? Are there any specific recommendations or tools for implementing this approach?
>
> Answer: Our next step is an empirical paper that is guided by this framework. We will conduct a meta-evaluation of LLaMa similar to the work of Zheng et al. 2023 [1], which uses language models as evaluators on different Machine Translation tasks, but increasing the power of the evaluator; we hypothesize that a learned evaluator increasingly distinguishes between LLaMa and samples from MT Bench as the complexity of the evaluator increases (measured by number of samples, and number of parameters).
>
> [1] Judging LLM-as-a-judge with MT-Bench and Chatbot Arena. Lianmin Zheng, Wei-Lin Chiang, Ying Sheng, Siyuan Zhuang, Zhanghao Wu, Yonghao Zhuang, Zi Lin, Zhuohan Li, Dacheng Li, Eric. P Xing, Hao Zhang, Joseph E. Gonzalez, Ion Stoica. arXiv, 2023.
>
> > What criteria would you propose for designing effective evaluators based on this framework? How would you prevent artifacts like evaluators favoring outputs closer to their training distribution?
>
> Answer: We note that in our definition, the evaluator is required to distinguish between the model and the "ground-truth". Therefore, if the evaluator overfits its samples (i.e., prefers its training distribution), it will end up rejecting even the ground-truth, which reduces its distinction. This initial paper is focused on introducing the framework and connecting it to existing literature, and this pressing question would be interesting to study in future work. We would like to highlight that the main point of our paper is to give our community a rigorous framework for stating and studying questions such as these, and so we are very happy to see this question arise already in this discussion!
>
> ### Regarding Missing References
> > This one may be relevant: Hutchinson et al. Evaluation Gaps in Machine Learning Practice. FAccT '22. https://dl.acm.org/doi/fullHtml/10.1145/3531146.3533233
>
> Answer: Thank you for the pointer! We will add this as related work.
>
> ### Regarding Typos Grammat Style And Presentation Improvements
> > 191: Does "they" refer to \mathcal{G}?
>
> Answer: Thanks for pointing out this typo - “They” here should be replaced by L_{G} and L_{E}.

---

### Meta-Review · Area_Chair_u9uo · 2023-09-15

**Recommendation:** 4

**Metareview:**

This paper received mixed scores of 4,3,2 (soundness) and 4,4,2 (excitement).

Strengths and weaknesses including the following were mentioned prominently:

Strengths:
- conceptual contribution to LLM evaluation (R1, R2, R3)
- well-written (R1)

Weaknesses:
- leaves unclear how the proposed framework can be implemented (R1, R2, R3)
- limited insights into relation to existing evaluation approaches (R1, R2)
- framework was criticized as too general to be useful (R3)
- may be reducible to PAC learnability (R3)

R3 provides a low soundness scores and has the following major concerns: (1) the work is very preliminary, (2) without concrete instantiation of evaluators, it remains unclear if the framework can be used for nontrivial theoretical analysis, (3) the  definition of pseudo-intelligence may be linked to PAC learnability.
I do not find (1), as explicated in the review, to be a major concern given the goals of this track.
The authors pushed back against the concerns (2) and (3), and I found their explanations to be convincing. In conclusion, I do not see major concerns against the soundness of this proposal for a theoretical framework for evaluation -- noting that it is certainly a first step, with substantial future work needed to work towards implementation.

---

### Decision · Program_Chairs · 2023-10-07

**Decision:**

Accept-Findings

**Comment:**

This paper received mixed scores of 4,3,2 (soundness) and 4,4,2 (excitement).

Strengths and weaknesses including the following were mentioned prominently:

Strengths:
- conceptual contribution to LLM evaluation (R1, R2, R3)
- well-written (R1)

Weaknesses:
- leaves unclear how the proposed framework can be implemented (R1, R2, R3)
- limited insights into relation to existing evaluation approaches (R1, R2)
- framework was criticized as too general to be useful (R3)
- may be reducible to PAC learnability (R3)

R3 provides a low soundness scores and has the following major concerns: (1) the work is very preliminary, (2) without concrete instantiation of evaluators, it remains unclear if the framework can be used for nontrivial theoretical analysis, (3) the  definition of pseudo-intelligence may be linked to PAC learnability.
I do not find (1), as explicated in the review, to be a major concern given the goals of this track.
The authors pushed back against the concerns (2) and (3), and I found their explanations to be convincing. In conclusion, I do not see major concerns against the soundness of this proposal for a theoretical framework for evaluation -- noting that it is certainly a first step, with substantial future work needed to work towards implementation.